# Iso- and Anisotropic Etching of Micro Nanofibrillated Cellulose Films by Sequential Oxygen and Nitrogen Gas Plasma Exposure for Tunable Wettability on Crystalline and Amorphous Regions

**DOI:** 10.3390/ma14133571

**Published:** 2021-06-25

**Authors:** Katarina Dimić-Mišić, Mirjana Kostić, Bratislav Obradović, Milorad Kuraica, Ana Kramar, Monireh Imani, Patrick Gane

**Affiliations:** 1Department of Bioproducts and Biosystems, School of Chemical Engineering, Aalto University, Aalto, 00076 Helsinki, Finland; monir.imani@aalto.fi (M.I.); patrick.gane@aalto.fi (P.G.); 2Faculty of Technology and Metallurgy, University of Belgrade, Karnegijeva 4, 11000 Belgrade, Serbia; kostic@tmf.bg.ac.rs (M.K.); ana.kramar@live.com (A.K.); 3Faculty of Physics, University of Belgrade, Studentski Trg 12, 11001 Belgrade, Serbia; obrat@ff.bg.ac.rs (B.O.); kuki@ff.bg.ac.rs (M.K.)

**Keywords:** micro nanofibrillated cellulose, surface plasma exposure of cellulose, wettability of cellulose film, plasma impact on surface cellulose structure, plasma-induced chemical changes on cellulose

## Abstract

The surface of cellulose films, obtained from micro nanofibrillated cellulose produced with different enzymatic pretreatment digestion times of refined pulp, was exposed to gas plasma, resulting in a range of surface chemical and morphological changes affecting the mechanical and surface interactional properties. The action of separate and dual exposure to oxygen and nitrogen cold dielectric barrier discharge plasma was studied with respect to the generation of roughness (confocal laser and atomic force microscopy), nanostructural and chemical changes on the cellulose film surface, and their combined effect on wettability. Elemental analysis showed that with longer enzymatic pretreatment time the wetting response was sensitive to the chemical and morphological changes induced by both plasma gases, but distinctly oxygen plasma was seen to induce much greater morphological change while nitrogen plasma contributed more to chemical modification of the film surface. In this novel study, it is shown that exposure to oxygen plasma, subsequently followed by exposure to nitrogen plasma, leads first to an increase in wetting, and second to more hydrophobic behaviour, thus improving, for example, suitability for printing using polar functional inks or providing film barrier properties, respectively.

## 1. Introduction

Cellulose, an abundant biopolymer, displays excellent physical and mechanical properties with potential for the production of high-end biobased composites. In lignocellulose, the cellulose molecular chains concatenate, forming strands stacked together as fibrils containing both crystalline and amorphous regions in nature, shielded overall by hemicellulose and lignin. These structures form the microfibrils and fibers that are connected through the intra and intermolecular hydrogen bonds [1]. There are numerous chemical and mechanical methods that can produce micro and nanofibrillated cellulose of different fibrillar morphologies [2] with common properties of being lightweight, exhibiting dimensions on the micrometre to nanometre scale and high specific surface area due to their high aspect ratio, supporting the provision of increased mechanical structural properties, of particular interest in composite materials [3,4,5].

Enzymatic pretreatment of pulp provides a first stage breakdown in an otherwise chemical-free route for producing low-charged micro nanofibrillated cellulose (MNFC), being an intermediate material between microfrillated cellulose (CMF) and nanofibrillated cellulose (CNF), in which microfibrils become surface nanofibrillated [6,7,8,9].

The use of such cellulose materials in areas as packaging often becomes restricted due to the water ad- and absorptive properties [9,10,11]. Surface modification is seen as a possible solution to this drawback, providing hybrid properties considered extremely valuable when considering functional printing applications, such as the printing of solar inks [12,13,14]. Different methods such as impregnation, layer-by-layer preparation, foam forming, curing, stretching upon drying and other processes have been used to try to improve usability of nanocellulose films, but, due to their delicate thickness and weight, they have not yet found industrial application, though advances have been made using calcium carbonate as a structural support agent [15,16,17]. Functional groups can also be introduced, further improving compatibility with a greater range of materials [18,19], including enhanced reactivity, for example, in developing nanophotonic substrates or materials with controlled wettability by various liquids [20,21,22,23]. One example of this wettability control is demonstrated when printing alternately oil-based and water-based functional inks [23,24,25]. Furthermore, the physical properties of a surface, such as roughness, also correlate with the final functionality derived from surface modification [26,27].

Direct modification of the surface of cellulose-based materials is frequently undertaken using techniques such as flame treatment or the deposition of metals. Other forms of irradiation are also employed, as well as corona discharge [28,29,30,31]. The latter two techniques can be considered examples of plasma exposure [32,33,34]. The chemical modification of a surface by grafting polymers onto the surface can also be achieved by an ionic mechanism, a coordination mechanism, coupling mechanism or free-radical mechanism [35]. Ionising radiation, including high-energy γ- and X-rays, as well as electron exposure, can be used additionally in specialised applications to cause surface etching/degradation, that affects the outer shell of the object and causes loss of its mechanical properties or hastens structural breakdown [19,36,37]. Such high-energy ion beams made from a single type of ion or combinations of many ionic species are already employed in modern technology, science and medicine, being particularly used to induce permanent modifications of the outer layers of substrates.

A plasma is a partially ionised gas that consists of positive and negative ions, electrons and radicals. It is often regarded as the fourth state of matter [38]. Despite being essentially chemical-free, it can, nonetheless, generate an activation potential at a surface. Plasma processing is widely used for surface modification of different materials because it is environmentally friendly, dry, and clean, often seen as its main advantage due to the absence of polluting and toxic chemicals [38]. At the substrate, the ion energy is controlled by the surface sheath potentials and the collisions that occur within the potential field [39]. The formation of a negative potential latterly prevents more electrons from settling on the surface and simultaneously attracts positive ions, such as protons, which are present from the action within the plasma and can result in surface attachment of chemical radicals [40].

Two contrasting plasma pressure conditions are commonly used in respect to the plasma chamber, namely, vacuum and atmospheric. Considering the thermodynamic states under which plasma can be generated, both equilibrium and non-equilibrium conditions are possible [41]. Non-equilibrium or cold plasma is characterised by a higher temperature of the electrons than is the case at thermodynamic equilibrium. Equilibrium plasma, or so-called high temperature or thermal plasma is characterised by an equal energy level for all species. Cold plasma discharges are also exemplified by microwave (MW), radiofrequency (RF) and dielectric barrier discharges (DBD) [28,38]. While MW and RF processes are usually electrodeless, with use of electromagnetic waves and antenna as transmitter, DBD plasma is generated in the region between two electrodes, which have at least one dielectric layer between them [20,27], with electrons, accelerated by the electric field, striking neutral gas molecules and resulting in the formation of highly reactive ions that cause chemical and topographical changes of the surface upon which they impinge [42,43]. Typically, the mean electron energy in DBD plasma is in the range of 1–10 eV [44,45]. In an open atmosphere, the plasma discharges can be produced with a gas flow between the electrodes [46,47,48]. At atmospheric pressure, DBD plasma can be used to modify or activate surfaces of a wide range of materials, from polymers and textile fibres to biological tissues [49]. In the textile, and paper and board industries, plasma exposure has found application in applying hydrophobisation, or hydrophilic coating, resulting in increased adhesion potential of the surface prior to applying a glue, as good wettability is a prerequisite for the adhesion of binding surfaces [49]. For such purposes, in relation to desired product, different gases are used: nitrogen/air (wet and dry), argon, and ozone plasma exposure [30,50,51,52,53]. These various interactions of plasma with surfaces lead to different plasma-based surface modification strategies [54,55]. While the plasma exposure mainly induces increased surface roughness by etching on the scale of micrometre dimensions, chemical reactions caused by plasma exposure are known also to produce even finer scale changes in surface roughness [56,57]. It has been documented that the combination of micro and macro roughness gives the best results to control wettability as they act either to decrease the depth of air pockets, whilst simultaneously lowering the contact angle to enhance super-wettability, or, on the contrary, create surface patterning, which leads to liquid meniscus discontinuity and hence super-hydrophobicity [58,59,60]. Cold DBD plasma exposure of MNFC-based composites can, therefore, be an alternative to conventional treatments that utilise fossil-based chemicals that increase roughness and compatibility with other molecules and particles, such as dyes, conductive fillers and inks [55]. Larger contact area, when coupled with chemical moieties from different gas types can lead to the enhanced application of ultrafine coatings [37]. In addition, if pulsed, this effect is exemplified when applying photovoltaic (PV) inks, as used for printing of solar panels and conductive smart films, comprising complex suspensions containing the organic electron acceptor (p-type component) and electron donor (n-type component) particles suspended in highly polar solvents and surfactant [61]. The electrolyte is highly polar, for example, and so sufficient wettability is needed by providing a polar surface, despite the parallel requirement for wettability by organic species [16,62]. This complex polar-dispersive surface energy balance is, therefore, critical for good printing results, i.e., drop spreading and drying [20,63].

Although the use of atmospheric pressure in DBD plasma eliminates the electrical discharge phenomenon within the chamber, the formation of non-equilibrium plasma at atmospheric pressure presents a challenge, as the discharge can easily contract into arcs, turning it into thermal plasma [56]. To overcome this sparking, special designs are implemented, for instance, DBD with reduced discharge gaps [57]. Different combinations of plasma conditions, such as power, pressure, and duty factor (time during which plasma generator can be used prior to the discharge breakdown per hour) will each favour one of these different pathways [23,57]. In this context, to achieve homogeneous treatment with a retention of functional groups, whilst avoiding cross-linking reactions, it is suggested that low wattage (discharge power) and short duty cycle can be advantageous [50] in ensuring that the original bulk of the sample remains intact. This can be achieved since the plasma species penetrate up to a maximum of 10 nm from the exposed sample surface, with chemical and morphological properties thus altered in the surface molecular layers only [45].

New findings have illustrated, in particular, the use of nitrogen plasma exposure of nanocellulose films to enhance wettability by a range of ionic solvents [54]. The effect of low-pressure oxygen and air plasma on the chemical structure of cellulose films and paper has also been frequently reported [18,30,50,64]. In previous research, we report an increase in wettability of MNFC films after nitrogen plasma exposure of films, which correlates with DBD control parameters’ pulsing frequency and power, and with the crystallinity of cellulose fibrils [44,55].

In this research reported here, we use exposure to plasma generated in oxygen, nitrogen and sequential combination in both gases to reveal the best route towards increased functional printability of MNFC films, whilst comparing with the untreated film mechanical and optical properties resulting from the intrinsic contact area of fibrils and rheological properties of their suspensions during film formation [33,43]. An additional novel aim of this study is to investigate the effect of enzymatic pretreatment prior to exposure to oxygen followed by nitrogen plasma. This reveals the contrast in response to plasma exposure from amorphous versus crystalline regions within films [55]. We identify a correlation between the observed change in surface free energy (SFE) of the MNFC films, before the plasma exposure, from the effect of enzymatic pretreatment thus exposing the relation between increase in nanofibril crystallinity with progressive enzymatic pretreatment and the impact this has on mechanical and barrier properties and primary surface roughness [65]. Measurements of the contact angle, and calculations of the SFE, surface roughness (atomic force microscopy (AFM) and optical laser confocal microscope (CLSM)) and material composition (X-ray photoelectron spectroscopy (XPS)) were also used to characterise the MNFC film surface before and after plasma exposure. Wetting properties of films change upon plasma exposure as a result of the film surface roughening, fibril length and amorphous domain areas. Rheological investigation of the MNFC cellulose suspensions was used to correlate change in fibril swelling and flocculation rate with the change of corresponding fibril to fibril contacts upon consolidation and hydrogen bonding during the film preparation process [66,67].

## 2. Materials and Methods

### 2.1. Preparation of MNFC

MNFC fibrils were produced from bleached hardwood Kraft pulp. Initially the pulp was washed with sodium hydroxide solution raising the pH to 10. Secondly, the suspension was washed with deionised water controlling to 8.2 µS conductivity. Enzymatic treatment employed ECOPULP^®^ R (Ecopulp Finland Oy, Koria, Finland). This enzyme is derived from a genetically modified strain of the fungus *Trichoderma reesei* [68,69]. Activity of the enzyme is reported as 17,700 nkat cm^−3^ cellulase retaining protein to a level of 93 mg cm^−3^ [29]. A quantity of 3 mg of enzyme per gram of pulp fibre was dosed to the final 2.5 *w/w*% fibre suspension. Hydrolysis was performed under continuous agitation at 57 °C and pH 5.5. Samples were collected at 0, 60, 150, 210 and 300 min. The enzymatic activity was terminated in each sample case by increasing the temperature to 90 °C and adjusting the pH to 9–10 using sodium carbonate. The samples were cooled overnight kept under cold storage. The suspensions were then individually refined using a homogeniser (model M-110P, Microfluidics, Norcross, GA, USA), operating at 2000 bar with a 100 µm restrictive flow gap. The MNFC suspensions thus produced were adjusted to a solids content of 1.65 *w/w*%.

Camera images in Figure 1 show the development of the gelation properties associated with the progressive enzymatic pretreatment of the bleached Kraft pulp in the production of the MNFC suspensions, revealing decrease in fibril size with respect to both loss of light scattering and increased surface water retention leading to a gel-like structure as the enzymatic hydrolysis is increased from 0 min to 300 min [69].

### 2.2. MNFC Suspension Characterisation

#### 2.2.1. Drainage

MNFC in suspension tends toward a gel-like structure with very high water retention. Therefore, to establish a relative drainage property a method was used whereby the impact of adding MNFC suspension into a starting pulp suspension is evaluated to indicate the effect MNFC has on the otherwise relatively free-draining bleached unrefined pulp. A DDA apparatus (Dynamic Drainage Analyser, Akribi Kemikonsulter AB, Sundsvall, Sweden) was used to determine the drainage properties of the pulp suspensions. The DDA device was operated under a constant vacuum of −0.10 bar, and the drainage was controlled through a stainless steel wire cloth with 37 μm opening diameter (400 mesh). Dilution of the pulp was made to a series of levels to form a number of host samples. The solids content of each, after subsequent addition of predetermined amounts of MNFC, could, therefore, be held constant at 3 g dm^−3^. The mixed samples were stirred for 1 min at 500 min^−1^ (rpm). Cationic starch (CS) was then added at 15 mg g^−1^ in relation to cellulose solids content into the suspensions to induce pulp fibre flocculation, in order to increase drainage to a high level, and mixed for a further 15 min. Into each of the preformed pulp suspension samples a known amount of MNFC cellulose equivalent ranging from 0 to 100 mg_MNFC_ g^−1^_pulp_ were added. After mixing for a further 15 min, stirring was stopped. The drainage valve on the DDA device was simultaneously opened, and time taken to drain was recorded. The MNFC suspension solids content was 2 g dm^−3^. Suspension volume for each test was 800 cm^3^.

#### 2.2.2. Water Retention

A similar method was applied to determine the water retention properties (water retention value (WRV)) of the fibril MNFC suspensions, once again due to the gel-like nature of the aqueous dispersion [67]. The WRV was measured in accordance with the SCAN-C 102XE standard, but using a slight modification, whereby 10 *w/w*% MNFC was added, as above for drainage, to form a series of ratios with a suspension of the bleached unrefined starting material pulp, and extrapolation of WRV to zero pulp was used to estimate the WRV of MNFC suspension alone [68]. Triplicate measurements were taken for each sample.

### 2.3. Rheological Evaluation of MNFC Suspensions

MNFC sample suspensions at 2 *w/w*% concentration were used for determining the rheological properties at 23 °C using an Anton Paar MCR 300 rheometer (Anton Paar GmbH, Anton Paar Strasse 20, 8054 Graz, Austria). Resulting data from the thixotropic gel-like materials were shown to have a variation within 10%. Noise in the raw data was reduced by applying Tikhonov regularisation [68,69]. Removal of obvious outliers was first undertaken (mostly manifest as negative values or those off scale). The appearance of unrealistic winding at the ends of the data chains was prevented by choosing only low order polynomials and by replacing the *y*-coordinate values of the end regions with ones calculated from these fittings. The smoothing regime could be applied also segmentally if needed.

#### 2.3.1. Shear Response

Steady state dynamic viscosity (*η*) reflects the shear flow properties. This was performed adopting bob-in-cup geometry. The “bob” in this case was a four-bladed vane spindle 10 mm in diameter and 8.8 mm in length. The outer metal cup was 17 mm in diameter. The flow curves were recorded as a function of reducing shear rate over the range $\dot{\gamma }$ = 1000–0.01 s^−1^, with a logarithmic spread of dwell time, advancing from 1 to 100 s.

#### 2.3.2. Viscoelastic Response

Due to the thixotropic nature of MNFC suspensions, oscillatory measurements are used for determination of rheological properties in the quasi viscoelastic region in which shear thinning complex viscosity (*η**) is recorded within the suspension viscoelastic stretching domain. Oscillatory viscoelastic measurements were made using plate-plate geometry (diameter 20 mm), adopting a serrated surface on both the upper and the lower plates. This was to avoid apparent wall-slip related to sample solids depletion. The gap setting was 2.3 mm. In addition, the bottom plate was contacted with a Peltier block to ensure temperature control, set to 23 °C. All suspensions were measured five times adopting a new sample each time.

To establish suspension homogeneity and the removal of regional structuration, the samples were presheared (shear rate, $\dot{\gamma }$ = 100 s^−1^) for 60 s and set to rest subsequently for 30 s. Amplitude sweeps were applied over a strain, *γ*, range between 0.1 and 500%, at constant oscillation angular frequency, *ω*, of 10 (rad) s^−1^, to reveal the region of quasi linear viscoelasticity and associated static yield stress, $\tau_{s}^{0}$. This was followed by a frequency sweep measurement spanning the range 0.01–100 (rad) s^−1^, held at constant strain (*γ* = 0.1%). The viscoelastic storage modulus, *G*′, and loss modulus, *G*″, were recorded as a function of strain amplitude and angular frequency. The complex viscosity, *η**, was evaluated from the data as a function of the increase in angular frequency (*ω* = 0.1–100 (rad) s^−1^).

#### 2.3.3. Yield Stress

For determining differences in the minimum stress required to induce flow in the MNFC suspensions, the static yield stress, $\tau_{s}^{0}$, was obtained from the viscoelastic region.

The dynamic yield stress, $\tau_{d}^{0}$, defined as the minimum level of stress required to maintain flow, was also determined, but now under shearing conditions using steady state flow curves [69].

The Herschel–Bulkley equation describes $\tau_{d}^{0}$ for shear thinning thixotropic fluids using dynamic viscosity, *η*, from the plot of the steady state flow curves,
(1)$$\tau_{d}=\tau_{d}^{0}+k{\dot{\gamma }}^{n}$$
where *k* is termed flow consistency and *n* the exponent in the power law expression, taking negative values for pseudo-plastic shear thinning.

As strain, *γ*, is constantly increasing at the constant angular frequency, *ω*, both storage modulus, *G*′, and loss modulus, *G*″, decrease upon instigation of viscous flow determined as the point at which deviation from linear elastic behaviour first occurs. The corresponding critical strain, *γ*_c_, occurring at the static yield stress, $\tau_{s}^{0}$, is given by
(2)$$\tau_{s}^{0}=G^{\prime }\gamma_{c}$$

Identifying differences in colloidal interactions, and effects of particle packing in relation to morphology of fibrils, can be achieved by forming log–log plots of the complex viscosity, *η**. These complex viscosity flow curves can also be fitted to a power law in a form equivalent of the Ostwald–de Waele model, applied by taking the root mean square of shear rate, ${\dot{\gamma }}_{\mathrm{rms}}$, recorded during oscillation,
(3)$$\eta^{*}=k^{*}{\left({\dot{\gamma }}_{\mathrm{rms}}\right)}^{n^{*}-1}=k^{*}{\left(\frac{\sqrt{2}}{\pi }\omega \right)}^{n^{*}-1}$$
where *k** and *n** are the complex viscosity magnitude equivalent of the dynamic consistency and power exponent, respectively. In the case of a Newtonian fluid the equation reduces to *n** = *n* = 1.

### 2.4. MNFC Film Preparation

The solid content of MNFC in suspension used during preparation of the films was in the range from 0.6 *w/w*% to 1.9 *w/w*%. Choosing this solids content range enabled the target film grammage of 60 g m^−2^ to be reached. The film production was made at 23 °C and 50% relative humidity (RH).

Films were formed using a sheet-former following the ISO standard 5269-1. Some modification of the screen, however, was made to aid fines retention, as previously reported [69]. This is needed because of the very strong water retention properties of MNFC, leading to possible loss of fine nanocellulose fibrils with drained water. The screen was therefore constructed using a polyamide monofilament fabric SEFAR NITEX^®^ 03-1/1 (Sefar AG, 9410 Heiden, Switzerland), having an open mesh pore size of 1 μm, which was placed on top of a supporting 125 μm metal screen. Double-sided adhesive tape, 5 mm wide, attached to the edges of the drying plate, was used to fix the film edge in order to prevent shrinkage during final air drying (Figure 2).

### 2.5. DBD Plasma Film Exposure

The DBD plasma device used in this work was home-made, as previously reported [43,54], and is shown schematically in Figure 3. The DBD assembly is held in a chamber with the respective gas injected into the discharge area at a flow rate of 6 dm^3^ min^−1^ by means of ten equidistant inlets around the perimeter, thus ensuring homogeneity of the gas flow. The MNFC film samples were exposed to the chosen gas plasma for 120 s and 240 s, respectively. The plasma generation took place under a discharge applied voltage of 6 kV, and 300 Hz AC. Higher voltage was avoided as it resulted in the thin MNFC films charring for the prescribed durations of time [45,46,47].

The combination of enzymatic pretreatment applied to the bleached Kraft starting material pulp and subsequent gas plasma type and exposure times for the manufactured MNFC films are shown in Table 1.

### 2.6. Material Characterisation

#### 2.6.1. Digital Imaging and Optical Microscopy

Digital optical images were collected using a Canon Eos 550C camera (Canon AG, Ōta, Tokyo, Japan). Images of samples were taken both with and without the aid of optical microscopy.

Microfibrillar content of the sample suspensions and films was studied using an Olympus BX 61 microscope (Olympus AG, Shinjuku, Tokyo, Japan).

#### 2.6.2. Scanning Electron Microscopy (SEM)

Scanning electron micrographs were obtained on a Zeiss Sigma VP SEM (Carl Zeiss AG, Oberkochen, Germany) using a secondary electron detector, with an accelerating voltage of 1.0 kV at a working distance of 5 mm, to characterise the morphology of the MNFC films. Samples were sputter coated using an Emitech K100X glow discharge apparatus (Quorum Technologies, Lewes, UK) equipped with an Ag/Pt electrode with a working current of 30 mA acting for 60 s, corresponding to a sputtered platinum layer of ~4 nm.

#### 2.6.3. Permeability—Air Flow Technique

Film permeability to air is measured according to the Tappi Test Method T 460 (Technical Association of the Pulp and Paper Industry, Atlanta, GA, USA) using the Gurley principle (Gurley Precision Instruments, Troy, NY, USA), which records the air resistance through ~6.45 cm^2^ circular area of the film using a pressure differential of 1.22 kPa. Results are quoted in μm (Pa s)^−1^.

#### 2.6.4. Surface Chemical Composition

Surface composition of the MNFC films prior to and following plasma exposure was analysed using X-ray photoelectron spectroscopy (XPS) (Kratos AXIS Ultra electron spectrometer (Kratos Analytical Ltd., Manchester, UK—a company of Shimadzu Corp., Nakagyo-ku, Kyōto, Japan)), applying monochromatic Al Kα irradiation at a power level of 100 W and adopting charge neutralisation. Samples were prepared under vacuum over a period of at least 12 h. Spectra were taken from a wide area survey for general elemental analysis. High resolution studies were also made to identify regions containing C1s and O1s, results being recorded from numerous locations. An in-situ reference taken from pure cellulose was also recorded for each investigated batch of samples. The analysis was conducted over a 1 mm^2^ area, and, XPS being a surface sensitive technique, the depth analysed was smaller than 10 nm. High resolution carbon data were fitted using CasaXPS open access software, adopting a Gaussian fit of four components tailored toward celluloses.

#### 2.6.5. Surface Roughness

##### Microroughness—Confocal Laser Microscope (CLSM)

The surface microroughness of the films was studied using a C TCS SP2 CLSM microscope (Leica Microsystems CMS GmbH, Mannheim, Germany). 750 μm × 750 μm images, generally suitable for biomacromolecules, were obtained specified at excitation and detection wavelengths of 488 nm and 500–530 nm, respectively. The image intensities, in both cross section and over the surface, were scanned in averaging mode maintaining constant conditions. The photomultiplier voltages were set at 700 V during surface, and 650 V for cross-section imaging, respectively.

##### Nanoroughness—Atomic Force Microscope (AFM)

Nanoscale roughness analysis was undertaken using atomic force microscopy, applying tapping mode (AFM, Dimension 3000, Bruker, Santa Barbara, CA, USA). Sample roughness was studied before and after exposure to plasma. No further image processing was used, apart from image levelling to remove effects arising from possible sample tilting. Scan sizes were typically of the order of 1 μm × 1 μm, and 5 μm × 5 μm.

##### MNFC Film Physical Testing

The density of the MNFC films, *ρ*_film_, was derived by recording their weight and respective volume, the latter being calculated knowing the thickness of the film, measured using a digital caliper, and the planar area, under equilibrium conditions at room temperature and humidity. The porosity (fraction of voidage in a material), *φ*_film_, can, therefore, be defined by the density ratio of the actual bulk density and the density as if it were all pure cellulose *ρ*_cellulose_, normally taken as 1460 kg m^−3^, using
(4)$$\varphi_{\mathrm{film}}=\frac{volume\;of\;voids}{total\;sample\;volume}=1-\frac{\rho_{\mathrm{film}}}{\rho_{\mathrm{cellulose}}}$$

Dynamic mechanical analysis was used to describe the mechanical properties of the films. The analyser (Model Q 800, APEM, Haverhill, MA, USA) was equipped with a 100 N load cell, operated at 5 mm min^−1^ crosshead speed following the American Society for Testing and Materials (ASTM) D 638-10 norm. Specimens were cut to a rectangular form, dimensions 50 mm × 0.53 mm, each having known thickness, using an Epilog Laser 35 W Zing scriber (Epilog Laser, Golden, CO, USA). At least eight specimens of each sample type were tested, and averaged values reported. Young’s modulus, *E*, was derived from the slope within the initial linear region of the stress–strain curve, expressed as the ratio of the tensile stress to axial elastic strain
(5)$$E=\frac{F/A}{\Delta L/L_{0}}=\frac{FL_{0}}{A\Delta L}$$
where *F* is the extensional force, *A* the cross-sectional area of the sample, and Δ*L*/*L*_0_ the fractional extension from starting length, *L*_0_.

### 2.7. Droplet Analysis of Surface Energy-Dependent Wetting

#### 2.7.1. Contact Angle and Free Surface Area

The contact geometry at the wetting front between a liquid, its vapour and the solid it is in contact with is defined in terms of contact angle (CA), *θ*, the cosine of which is determined by the wetting force, generated by the liquid contacting the solid, resolved along the planar surface direction, *σ*_LV_cos*θ*, where *σ*_LV_ is the parameter of surface tension between the liquid and its vapour (formed in contact with air in the typical case) (Figure 4). The surface wettability is an important topic of study as it plays the leading role in surface adhesion, coating and printing phenomena [70,71]. The CA at the liquid–solid–air interface, *θ*, distinguishing between surface hydrophilicity and hydrophobicity, is frequently studied by applying a droplet of the chosen liquid onto the surface. Changes in the droplet behaviour can also be studied by following droplet shape. If the surface is completely hydrophobic, for example, a droplet of water forms a perfect sphere in air. In intermediary cases, from partial hydrophilicity onwards to complete wetting the droplet shape changes, which allows for minimising the surface energy between the liquid and the solid [63]. When droplets are placed on a surface, forces acting on the drop have to meet equilibrium represented by Young’s Equation,
(6)$$\sigma_{\mathrm{LV}}\cos\theta =\sigma_{\mathrm{SV}}-\sigma_{\mathrm{SL}}$$
where *σ*_SV_ and *σ*_SL_ are the respective surface energies of the interfaces defined by solid–vapour and solid–liquid, respectively, and in which the behaviour for water is classified as being hydrophilic in the range *θ* < 90° or hydrophobic if *θ* > 90°.

For hydrophilic properties, during wetting, the liquid acts to fill the roughened surface making contact with the entire surface, while for a hydrophobic surface wetting contact becomes limited to the upper surface parts of the roughness (liquid–solid) only, with air remaining in the pockets/voids of the surface (liquid–air) [52,71] (Figure 1) (inserts). Thus, the roughness increases effective wetting area, as the roughened contact line with the liquid is longer than the equivalent line length on a perfectly smooth surface, which, in turn, increases the work necessary to move the drop along the plane of the surface. On the one hand, strong liquid wetting by water is driven by the high solid surface energy, *σ*_SV_, when hydrophilic. The hydrophobic nature of a surface, on the other hand, is derived mainly from the combination of sufficient surface roughness having low surface energy. These two situations, hydrophilic and hydrophobic, respectively are described in turn by the Wenzel (Equation (7)) and Cassie–Baxter models (Equation (8)), the former being specifically limited to non-porous media only,
(7)$$\cos\theta^{*}=r\cos\theta$$
(8)$$\cos\theta^{*}=r_{f}\cos\theta +f-1$$
where *θ** represents the apparent (observed) CA, and *θ* is the true microscopic CA defined by the Young’s equilibrium. In the Wenzel case, *r* is the roughness ratio between the actual and planar-projected continuous surface area, where *r* = 1 for a smooth surface, and *r* > 1 for a rough surface. For the hydrophobic Cassie–Baxter case, *r_f_* is the roughness ratio with respect to the wetted surface area in relation to the planar-projected surface area, where *f* is the fraction of total solid surface area only that is wetted by the liquid. In the case where *f* = 1 and *r_f_* = r, the Cassie–Baxter relation naturally reduces to the Wenzel equation.

In extreme cases, the roughness factor can contribute even more significantly than the Cassie–Baxter model can reliably predict. In the limit, {*r_f_*, *f*} → 0 to obtain super-hydrophobicity, i.e., when water on the surface exhibits a contact angle greater than 150°, and there is resulting extremely low adhesion of a liquid to the surface, as a result of which a drop can slide easily over and off the material surface.

CA, and so surface free energy (SFE) of the MNFC films derived therefrom, were recorded using a video camera (CAM 200) attached to an optical contact angle meter (KSV Instruments Ltd., Espoo, Finland). A droplet of chosen liquid, up to 6.2 µL, was automatically dispensed from a microsyringe onto the film surface. Upon touching the film surface, the image of the droplet was recorded and analysed by the incorporated software (CAM 2008). The identification of contact line geometry and respective evaluation of the CA is thus provided Figure 5. For visual effects, reported here using digital optical camera images, the water was dyed using Eosin dye 0.5% (Merck Life Science OY, Keilaranta 6, 02150 Espoo, Finland). To illustrate enhanced wetting, a mix of water and isopropanol was used (25%) to reduce liquid surface tension, σ_LV_.

The average receding contact angle, *θ*_r_, observed from droplets of each of four pure different surface tension probing liquids (water, formamide, diiodomethane, and ethylene glycol) was recorded. Five separate analyses on each sample were averaged, and the surface energy components (dispersive Lifschitz–van der Waals, *σ*_LW_, and acid-base polar, *σ*_AB_, and total solid surface energy, *σ*_tot_) were calculated. The total surface free energy, *σ*_tot_, is the value reported in this work, avoiding any controversy regarding applicability of the separate energy model for heterogeneous surfaces.

#### 2.7.2. Printing

Functional printability is an important end-use criterion for micro and nanocellulose films [72,73,74]. The printability on the film surface, prior to and after plasma exposure, was exemplified using a functional inkjet ink, in this case a photovoltaic (PV) inkjet-printing ink. PV inks comprise a complex formulation of materials, solvent and surfactants which act to keep the p-type and n-type components fully de-mixed, as reported previously [16,74]. The ink was dispensed using a piezoelectric laboratory-scale drop-on-demand (DoD) inkjet printer (Dimatix 2831-DMP) [74]. The ink diluent solvent used to prepare suitable ink viscosity was 3-methoxypropionitrile (Sigma Aldrich Corp., St. Louis, MO, USA).

## 3. Results and Discussion

### 3.1. Dewatering and Rheological Evaluation of Suspensions

The values of dewatering, water retention, consistency coefficients, *k* and *k**, and the power indices, *n* and *n**, are shown in Table 2. All measurements were repeated five times, and data variation was within 10%. Both consistency and shear thinning are seen to decrease as a function of enzymatic pretreatment time.

It is clear to see from Table 2 that dewatering rate decreases with increasing enzymatic hydrolysis time, as fibrils become thinner and finer. The suspensions become more gel-like [69]. At the same time, the crystalline fraction of fibrils necessarily increases and so, too, the water trapping structure/flocculation within the sample. However, once the structure is broken, the fibrils display increased mobility within the flow [3]. The effect seen as increase in shear thinning behaviour, together with decrease in consistency in both viscoelastic and steady state dynamic viscosity curves, is at least partly resulting from the combination of gel-like properties and less entanglement of the more crystalline shorter fibrils. Both static, $\tau_{s}^{0}$, and dynamic yield stress, $\tau_{d}^{0}$, increase as the suspensions become more and more gel-like. This reflects the floc structure breakage requirement in suspension when fibrils are set in motion. More enzymatically hydrolysed fibrils being smaller and more crystalline results in them moving more freely in the flow direction [3,69,75].

The viscoelastic rheological response curves of the MNFC suspensions, showing the change in viscoelastic response as a function of strain and frequency of oscillation, are presented in Figure 6. In addition to the effects revealed by the shear thinning coefficients, there appears a region of secondary aggregation at high shear rate occurring after the primary shear-thinning domain, the effect being more marked for the longer fibre samples having zero or less enzymatic pretreatment. It is considered that this aggregation is driven by entanglement of longer fibrils.

### 3.2. Morphological and Mechanical Evaluation of MNFC Films

Digital optical camera images of the films prior to plasma exposure are presented in Figure 7, together with respective SEM images. The films show distinct differences in transparency and uniformity as a function of enzymatic hydrolysis time. SEM images show the evident presence of fibres, and the progressive filling of void spaces between these microfibril flocs by the dense uniform structure of the fine nanofibrils very tightly associated together, particularly so after 210 min and 300 min hydrolysis time. After 150 min and greater, the films have much higher transparency but still visible flocculation in some areas, and network formation of broken microfibrils. Relatively high transparency is reached after 210 min with only minor flocculation (more clearly seen at the left side of the sample image), what small flocs there are have become packed into the dense nanostructure with very low apparent surface roughness. Finally, at 300 min of enzymatic pretreatment, the resulting film is effectively fully transparent, without visible flocculation. The surface is also very smooth with uniform packing of the fibrils packed into a continuous web with a highly reflecting surface.

Mechanical and light scattering properties of the MNFC films are shown in Figure 8a, which reveals that light scattering decline is a natural corollary of the increasing fibril fineness and, hence, increased density. This finding supports the interpretation that decreasing light scattering in these systems is related to the loss of voids between fibrils rather than the low refractive index of the material itself.

In Figure 8b, an interesting correlation is also seen between the dried film mechanical elastic modulus and the behaviour of the relevant suspension in the prior wet state. The elastic Young’s modulus is proportional to the water retention in the suspension, as shown by the WRV, and inversely proportional to the minimum dynamic stress needed to keep the system flowing. This correlation reveals the strength property to be a function of the production of nanofibrils, as to be expected, having an influence on flow behaviour, Also, as the density increases, correspondingly the permeability of the film, measured independently using the air flow technique, rapidly becomes very low, ranging from ~70 μm (Pa s)^−1^ without enzymatic pretreatment to ~1 μm (Pa s)^−1^ after 60 min pretreatment, and thereafter immeasurably small.

### 3.3. Hierarchical Roughness and Structure Regimes

As was seen from the micrographs in Figure 7, the basic level of surface morphology control is derived from the enzymatic hydrolysis pretreatment of the fibres prior to MNFC mechanical production and film formation. The longer the enzymatic pretreatment time the more homogeneous the surface becomes. It is, therefore, valuable to quantify the modified properties of the sample surface, firstly, in terms of roughness profile, and, secondly, following plasma exposure, the impact of surface etching and chemical moiety modification. Surface roughness topology is itself split into surface-distributed formation microroughness and textural nanoscale features, related to the physicochemical nature of the surface boundary and species interface material. Thus, in total, there are two main experimental levels of material hierarchical structure, namely the starting formation homogeneity, with two sublevels of microroughness and nanotexture, and plasma-induced changes, comprising the two sublevels of etching and chemical modification, i.e., four parameter sets to study.

### 3.4. Microroughness Determined by CLSM Scanning

Figure 9, showing the CLSM microscan and profile height distribution as insert, illustrates the progression of surface impact of the N plasma at two exposure times, (i) 120 s and (ii) 240 s, on the basic set of MNFC films derived from (a) 0 min, (b) 60 min, and (c) 210 min enzymatic pretreatment. Similarly, in Figure 10, the scans after oxygen plasma exposure are illustrated for (a) 120 s and (b) 240 s exposure for MNFC films made after the range of enzymatic pretreatment times comprising 0 min, 60 min, 150 min and 300 min. The root mean square (RMS) microroughness in each case is shown in Table 3.

From Table 3, and Figure 8 and Figure 9, the effect of longer plasma exposure for both nitrogen (N) and oxygen (O) discharge is to increase the surface microroughness progressively. For each individual MNFC film, the overall boundary levels of roughness are dependent on the starting roughness of the film prior to plasma exposure, i.e., the longer plasma exposure time approximately raises the roughness level of the smoother highly enzymatically pretreated material to that of the non-exposed enzyme unpretreated sample. This interesting finding suggests that plasma etching probably occurs on a similar dimension scale to that of the MNFC basic building block structure, i.e., the cellulose crystalline versus amorphous component dimension, which is the same scale at which the enzymatic hydrolysis acts, progressively separating out the amorphous component. Thus, breakdown of the amorphous constituent of the film results in more uniform topology and greater resistance to plasma discharge. We may also deduce at this size scale that limiting single gas plasma exposure to 120 s is a good compromise in respect to surface profile, and it is on this basis that the combination concept of O plasma followed by N plasma is proposed, each step being limited to 120 s, respectively.

The complex description of the interplay between improved homogeneity/micro-smoothness (inverse CLSM roughness) of film formation as enzymatic pretreatment is increased and the counter action of plasma exposure is captured in the rotatable 3D interpolated surface plots in Figure 11. The surfaces are shown from two perspectives to highlight the two independent variables of pretreament hydrolysis versus plasma exposure. Interestingly, the more aggressive etching effect of O plasma can be clearly discerned using this display technique.

Nanoscale surface texture is revealed by AFM tapping mode analysis. Colour contour AFM surface profile plots from MNFC films, collected prior to and as a result of plasma exposure, reveal the nanoscale surface texture, or rugosity, generated progressively after O plasma exposure for 120 s, followed by N sequentially, 240 s total, respectively (Figure 11, Figure 12 and Figure 13).

As was seen in the CLSM microroughness analysis (Table 3), the AFM texture nanoscale roughness prior to plasma exposure is also initially inversely dependent on the length of enzymatic pretreatment time (Figure 12), with the surface texture becoming clearly progressively smoother, ignoring microscale undulations, as the pretreatment time extends from 0 through 300 min. The contour maps for 60 through 300 min (Figure 12b–d) pretreatment indicate that there are voids present of between 1 and 2 µm surface opening width, which gradually disappear in the film formation step as enzymatic pretreatment extends to 300 min, and the surface becomes flatter with less voids, all of much smaller size [59]. This means that the degree of enzymatic hydrolysis results directly in observably increasing nanoscale texture smoothness enacted by shortening the pulp fibre component chains as they subsequently separate under the mechanical homogenising. The crystalline parts eventually make up the resulting film formed surface [44,48].

Depending on the breakdown by O plasma of retained amorphous constituent regions in the film surface, the distribution of surface nanoscale texture after plasma exposure is determined. Thus, as the retained amorphous constituent is reduced, as a function of enzymatic pretreatment, so the distribution of texture generation becomes more isotropic, whilst patchy amorphous retention, characteristic of less enzymatic pretreatment, results in anisotropic plasma etching—compare, for example, Figure 13a, showing anisotropic texture, with the highly isotropically textured rugosity of Figure 13c. The latter is related to the uniform strong etching by the plasma of the more single component crystalline cellulose domains in the film surface. These results confirm the earlier desire to reach a compromise on the degree of O plasma etching. Thus, clearly, 240 s of O plasma exposure prior to N plasma exposure would create a surface that is too highly etched. For example, the smooth starting film after 300 min enzymatic pretreatment becomes highly, albeit uniformly, etched by the O plasma, creating both nanoscale texture as well as microscale indentations.

The compromise of using 120 s O plasma exposure followed by a further 120 s N plasma exposure is clearly justified when viewing Figure 14, where surface texture becomes highly defined but on a fine scale with optimal combination of enzymatic pretreatment and controlled exposure during the sequencing of O + N plasma exposure.

Correlation between the hierarchical microroughness, determined by CLSM, and nanoscale texture, shown by AFM, is presented as comparable surface net plots in Figure 15. The naturally lower level of nanoscale roughness (AFM) is displayed, together with the clarity of the counteracting relationship of the O plasma etching causing roughening versus the original formation improving effect of enzymatic pretreatment of the fibre feed prior to MNFC production.

### 3.5. Contact Angle and Surface Free Energy (SFE)

Wettability was assessed from measurement of contact angle (CA). As was expected, the increasingly fine and more crystalline MNFC fibrils formed out of the longer enzymatic pretreatment of the feed fibres led to a progressive decrease in wettability by water as the crystalline cellulose revealed its drop in polar surface energy component. The situation changes when the surface becomes exposed to N plasma, which induces an increase in wettability for water drops, the increase being most pronounced for the more crystalline samples [76]. However, as the roughness introduced with longer exposure time of the less crystalline (shorter enzymatic pretreatment) to both N and, especially, O plasma is seen to act to decrease wettability by water. The initial etching by N plasma is accompanied by N-based radical attachment, whereas with O plasma exposure etching is more pronounced at the amorphous-rich areas of the surface parts of fibrils.

Table 4 contains the results of SFE, *σ*_tot_, which reflect the case for water wettability observations reported above. We see that breakdown, firstly, of the amorphous content, resulting in debonding and, hence, roughening [63], is, secondly, replaced by the action of N-based radicals and their chemical reaction with the cellulose fibril surface, such that the higher average SFE values are once more regained on the now more crystalline samples following N plasma exposure. This occurs both when exposed to N plasma solely, or when combined with prior O plasma exposure, resulting in SFE values greater than the theoretical 59.4 mJ m^−2^ for cellulose alone. Extended exposure to O plasma for 120 s and 240 s led to an increased O:C ratio in all samples (discussed further later in relation to elemental analysis with XPS method). The surface energy increase in the case of the combined O followed by N plasma nonetheless still occurs despite the fact that the O:C ratio increases during the step prior to O plasma exposure, which alone, without the subsequent attachment of N-based radicals, leads to a decrease in free surface energy for all samples except the 150 min enzymatic pretreated sample, which is an inexplicable exception where the SFE remained unchanged.

SFE observations reveal that a higher amount of polar component upon plasma exposure is present than for untreated MNFC films, and a plasma-induced increase in polar component by the N adsorption acts to promote the compatibility with the highly polar liquids. The images in Figure 16a–c support the numbers given in Table 4, where increased wetting of the MNFC film surface by drops of water, dyed with Eosin dye 0.5%, as a function of plasma exposure time is paralleled by the greater pick-up (trapping) of printed ink colorant [64].

The surface tension of water is reduced by the addition of 25% isopropanol enabling instantaneous spread of droplet over the entire film surface, resulting in curling of the film (Figure 17). This “super-wetting” effect is mostly evident for the film made from fibrils that underwent the longest enzymatic pretreatment hydrolysis time, for which, prior to plasma exposure, wetting was the least. Reduction of the amorphous parts of MNFC fibrils with progressive hydrolysis time is responsible for the hydrophobic tendency arising from the more crystalline fibrils; however, after surface etching on exposure to plasma, and radical deposition on the roughened surface, wettability is recovered.

### 3.6. Elemental Analysis of Plasma Treated Films (XPS)—Impact on SFE

The surface chemical species are revealed in the XPS spectra, from which the atomic concentration (%) of C-C, C-O, O-C=O, O:C ratio and N can be derived. The effect of surface modification resulting from plasma exposure can be seen in Figure 18. The plots in Figure 18 display the effect of enzymatic pretreatment and plasma exposure on the sample SFE, shown on the *y*-axis. The contribution of the relative chemical species is represented by the atomic concentration determined by XPS along the independent variable *x*-axis. The data are distributed according to the sample enzymatic pretreatment and plasma exposure histories, with each labelled sample explored marked via the horizontal dotted line recording its given SFE. The joining lines, guiding the eye over the plot, are used to indicate the change resulting during each experimental history step.

It can be seen from Figure 18 that the level of attachment of N- and O-based functionality increases as a function of the exposure time, and increases more for films having had progressive removal of amorphous content by enzymatic pretreatment of the feed fibres [76]. The samples with increased crystalline proportion, resulting from longer enzymatic pretreatment, nonetheless show similar C-C bond content to those samples having experienced less hydrolysis time. However, with the simultaneous reduction of the amorphous part with increased pretreatment enzymatic hydrolysis, the number of C-O groups is seen to decrease while C=O groups and other C and N containing groups are formed.

Total O is increased, but on account of carbonyl groups (C=O), which are less polar than OH and COOH, and their increase serves as an indicator for the more hydrophobic behaviour. O plasma exposure led to the increase of SFE above the starting value of unexposed films, the highest being in the case of 150 min and 210 min enzyme pretreated films, with O:C ratio being the largest on 210 min fibril pretreated films exposed for 240 s to O plasma.

Surface chemistry is not changed significantly, meaning that, in most of the samples, the magnitude of the O1 s peak remains the same, although there is a decrease of O1s on the films produced from MNFC pretreated with enzyme for 150 min and 300 min during the first 120 s of plasma exposure, and then an increase after 240 s plasma exposure, but not above the starting unexposed values. In deconvoluted spectra all peaks corresponding to O containing groups decrease after O plasma exposure, except for the signal corresponding to COOH groups, meaning that the etching effect is very pronounced but, concurrently, oxidation of the OH groups on the surface of crystallites occurs. When consecutive plasma exposure is performed on the film sample made from non-pretreated fibrils, first in O and second in N plasma, the SFE increases by 6.8% [72,76,77]. In this case of no enzymatic pretreatment, both plasma exposure steps increase the SFE even though total O1s decreased along with the increase in C-C. In addition, presence of OH groups is smaller by almost 50%, as well as for C=O, but a very large increase was observed in a signal corresponding to COOH groups, being nine times higher compared with the unexposed sample, the biggest change amongst all the samples investigated. Regarding the surface roughness, microroughness decreased, as well to a lesser degree nano surface texture roughness [55]. The synergistic effect of both plasmas provided an etching of amorphous areas and the hydroxyl groups associated with them, but new more polar carboxyl COOH groups are introduced which provide for better water wettability, increasing hydrophilicity whilst decreasing hydrophobicity.

The combined influence of surface chemical species, generated by plasma-induced modification, and specific roughening by surface etching, creates the dependent change in effective SFE as derived from liquid CA [29]. This dependency is presented in Figure 18a, showing the impact of plasma exposure on non-enzymatically pretreated derived MNFC film, where the action of the various plasma exposure leads to a chemical species concentration-driven change of SFE (shown by the gradient line change arrows as a function of atomic concentration). For intermediate enzymatic pretreatment, at 150 min (Figure 18b) and 210 min (Figure 18c), the transition from O plasma etching impact on SFE toward a control via plasma type and combination is seen. Finally, at 300 min (Figure 18d), enzymatically treated MNFC films show the greater impact of plasma type on the most crystalline cellulose film. It is, therefore, evident that chemical composition of the surface changes can be understood also by following the connecting guidance lines in Figure 18, being more altered as a result of N rather than O plasma exposure.

Surface voidage is the major result of O plasma exposure in the coarser particulate systems, as previously described, due to effects of charge generation, fibril debonding etc. [77]. Combined exposure of O followed by N plasma equilibrates between the etching effect and more uniform deposition of N ions as radicals on the roughened surface.

### 3.7. DoD Printing on DBD Plasma-Exposed Films

Until this point in the discussion, the overall application benefit of dual plasma exposure, O plasma followed by N plasma, has only been alluded to, but not clearly stated. The increasing contribution of the micro and nano texture roughness induced by plasma etching, primarily by O plasma, and the related oxidative species deposition, followed by the polar component surface modification introduced by the cationic N adsorption under the presence of N-based radicals induced by N plasma, tunes wettability for various liquids on the MNFC films [55,56,78]. Printability, for example, is key to achieving high performance from functional inks [33,79]. Even minor changes in uniformity of printout can be considered significant in the context of hydrophilicity and morphological changes in the functional printed layer.

To illustrate the achievable change in printability, a simple inkjet print is made using a highly polar original yellow-orange tinted functional ink (Figure 19). It can be seen clearly that the shade morphs from light yellow to darker orange as the enzymatic pretreatment and plasma exposure sequence and time are extended, the latter finally matching the original orange colour of the inkjet ink. In addition, the print uniformity is clearly improved, in this case beyond that already shown by Dimic-Misic et al. [55], achieved by N plasma alone, by using the combination of O followed by N plasma on extended enzymatically pretreated MNFC film.

The importance of “shade” and uniformity is not confined to colour image printing but is critical to the function of light interacting dyes. For example, solar inks must retain their wavelength sorbance criteria. Using this technique of substrate preconditioning means that not only can an ink be optimised in manufacture but can also be optimised for optical performance during the printing step depending on the substrate chemical and micro/nanoroughness state.

While preventing excessive wetting, and tuning hydrophilic-hydrophobic domains of films by playing with isotropic etching and chemical moiety modification using sequential gas plasma exposure, the use of thin micro and nanocellulose films and coatings as a substrate can be optimised for various degrees of liquid philicity, ranging from highly polar anionic or cationic inks, for example [65], to oil or emulsion based inks.

## 4. Conclusions

Combining the findings presented in this work, Figure 20 provides an illustrative schematic demonstrating from Figure 20a through Figure 20c the stage by stage impacts of the various experimental steps. The impact on morphology of the MNFC fibrils, resulting from the degree of enzymatic pretreatment, in turn defining the basic surface structure of the films, is shown in Figure 20a, and the effect it has on water droplet wetting is illustrated. The effect of O plasma is represented in Figure 20b, having a strong etching effect on the film surface, differentiating between crystalline and amorphous domains along the cellulose fibrils, once again impacting on water droplet interaction. Finally, Figure 20c illustrates the combination of surface changes by firstly etching with oxygen plasma, and secondly exposing to nitrogen plasma, the latter acting to change the etched surface chemically and so providing the subsequent desired control of wetting properties.

The key point to conclude is related to the mix of changes that can be achieved between chemical surface moieties combined with the differential etching of amorphous versus crystalline cellulose components. For example, firstly, by reducing the amorphous content of cellulose from the outset, by applying enzymatic hydrolysis to the fibre feed used to form a micro nanofibrillated cellulose (MNFC) film, subsequent oxygen (O) plasma etching becomes more uniform, and so a controlled homogeneous roughness texture is formed. In the same example of O plasma exposure, secondly, chemical oxidation species are formed which contribute to a nanoscale patchwork of oleophilic (lipophilic) sites. Thirdly, a subsequent exposure to N plasma regenerates a similar patchwork of hydrophilic sites. This duality leads to an amphiphilic surface, which can be tuned over a wide range of desired wettability, i.e., since microstructural and nano texture roughness also leads to an extended liquid–solid contact line, then tuning surface energy simultaneously by selective plasma-induced cellulose filament structural modification can provide the opportunity to change surface wettability dramatically, even, in principle, to the level of inducing super-hydrophilicity or super-hydrophobicity. Inducing, on the contrary, amphiphobicity is in principle also possible using the techniques reported here, for example, reversing sequential, or preselecting single, plasma use.

Liquid phobicity is desirable to meet the needs of a broader range of applications, for which the other main properties of cellulose films, such as strength, gas impermeability and oil barrier are important, so as to include, newly, liquid repellence in fields such as printed media, liquid packaging, printable barrier films and textiles.

Adopting microscopic studies, including optical, electron and atomic force microscopy, with detailed contact angle surface energy measurements, enables analysis of the complementary contribution of apparent surface energy and surface micro and nano texture roughness.

## Figures and Tables

**Figure 1 materials-14-03571-f001:**
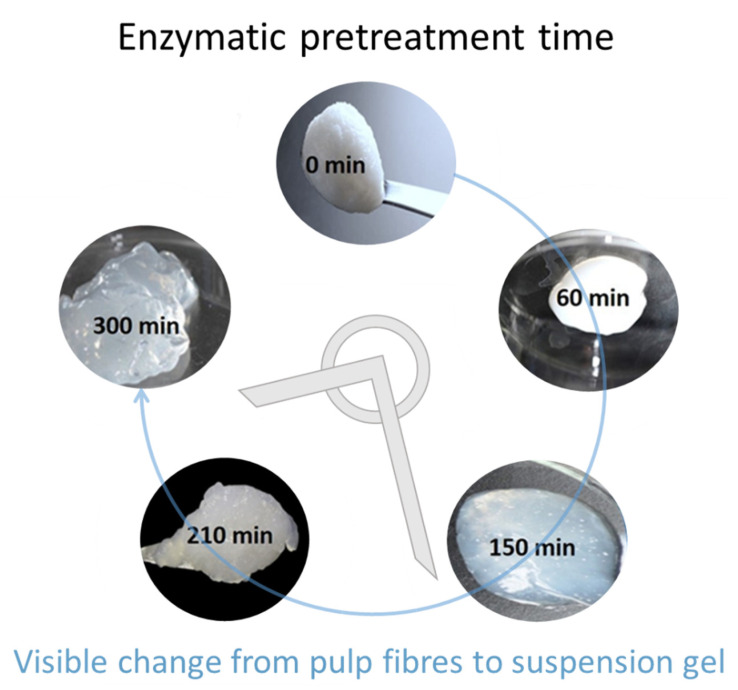
Images of the obtained fibril samples showing impact of enzymatic pulp pretreatment time leading to conversion from a light scattering fibre pulp to the translucent gel nature of the respective MNFC suspensions produced.

**Figure 2 materials-14-03571-f002:**
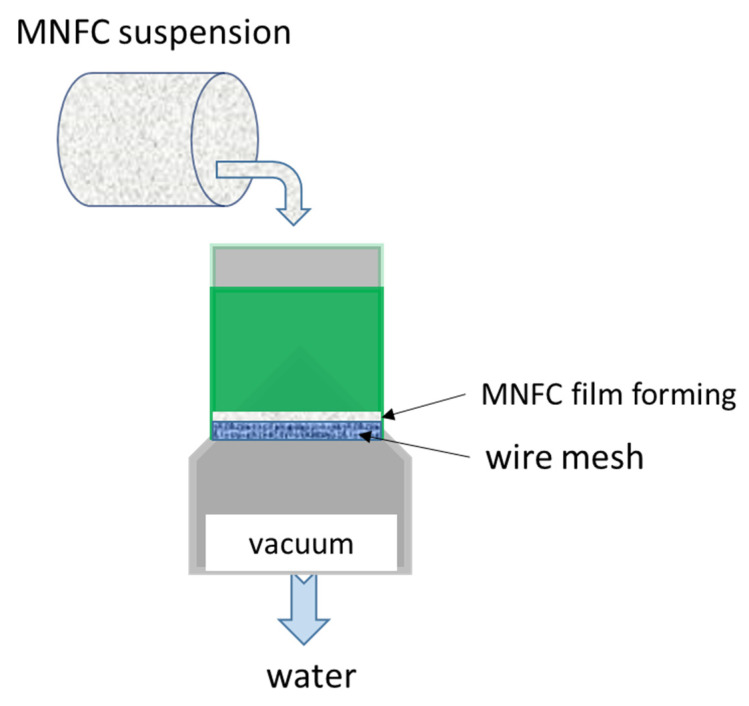
Schematic of MNFC film preparation on a laboratory sheet former adopting a polyamide filament screen put over the 125 µm mesh to prevent nanofibrils draining with flushed water.

**Figure 3 materials-14-03571-f003:**
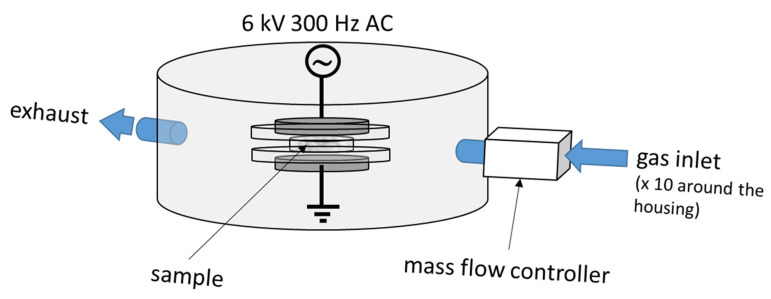
Schematic illustration of the DBD device, showing the two electrodes with the sample between them.

**Figure 4 materials-14-03571-f004:**
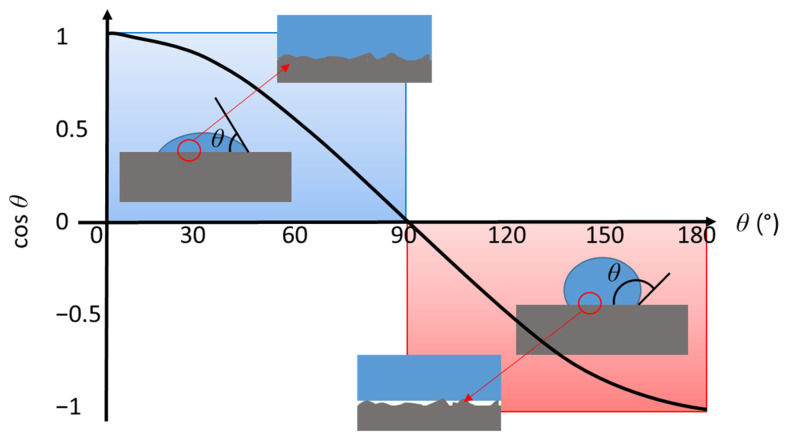
Schematic illustration of the increase in contact angle *θ* between hydrophilic and hydrophobic surfaces: the inserts show the difference between the cases when there is also a surface roughness present, whereby the surface becomes liquid filled in the case of perfect wetting versus the retention of air in the surface voids in the hydrophobic case.

**Figure 5 materials-14-03571-f005:**
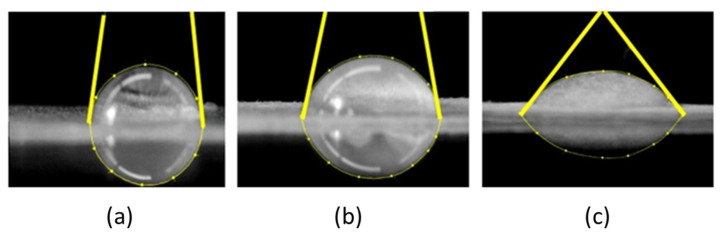
Image analysis (KSV Instruments Ltd.) identification of the droplet contact and circumference points of a droplet on MNFC film derived from 300 min enzymatic fibril pretreatment: (**a**) high contact angle, (**b**) medium contact angle, and (**c**) low contact angle.

**Figure 6 materials-14-03571-f006:**
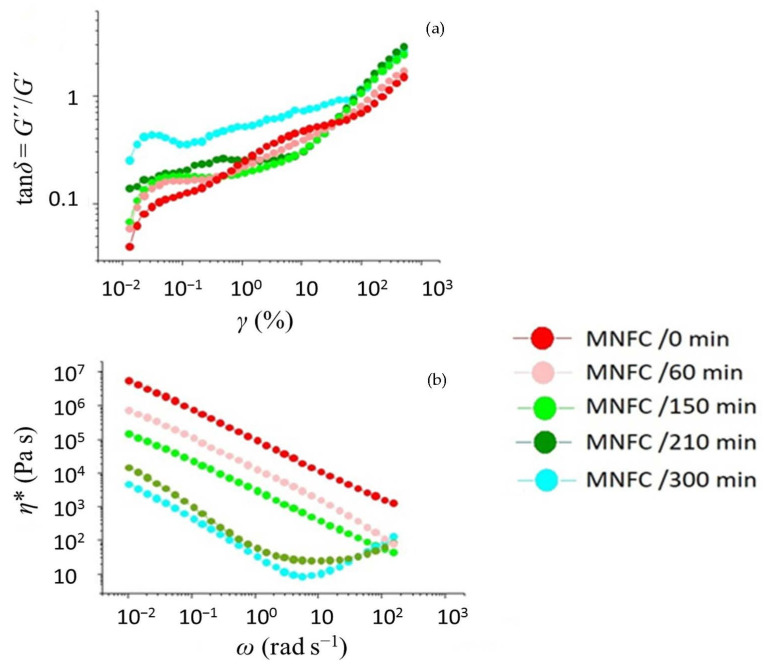
Flow curves of MNFC suspensions showing the progressive effect of enzymatic pretreatment time: (**a**) phase response, tan *δ*, to strain, *γ*, (**b**) complex viscosity, *η**, response to angular frequency, *ω*.

**Figure 7 materials-14-03571-f007:**
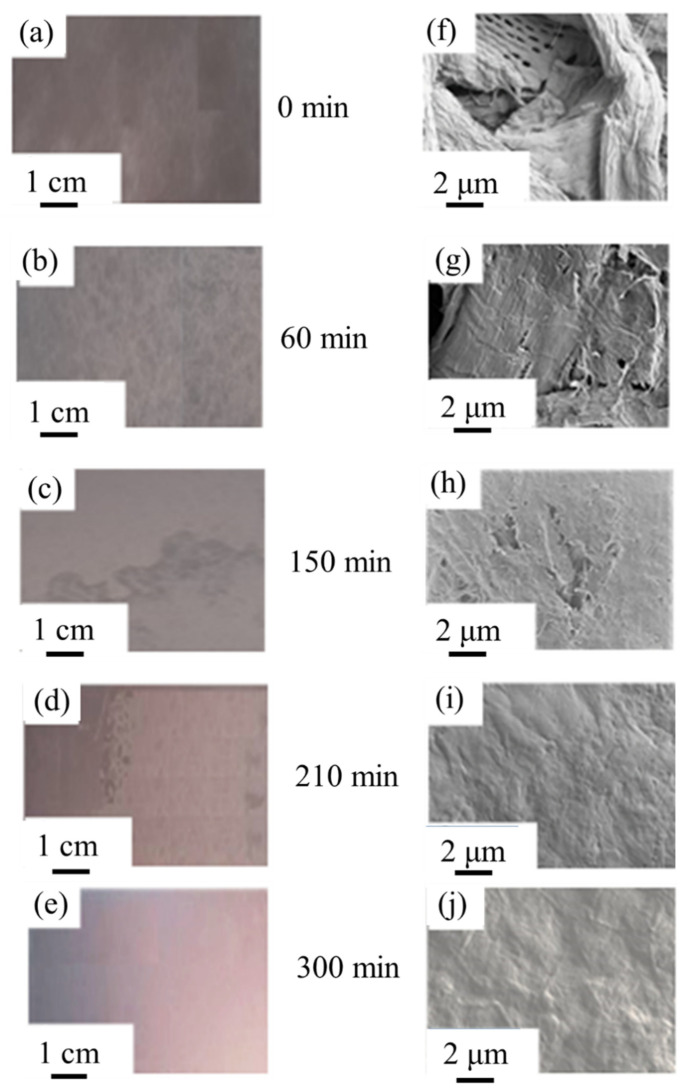
MNFC film images obtained with digital optical camera (**a**–**e**), and SEM (**f**–**j**) showing the increase in uniformity as fibres are progressively broken down as a function of enzymatic hydrolysis pretreatment over time.

**Figure 8 materials-14-03571-f008:**
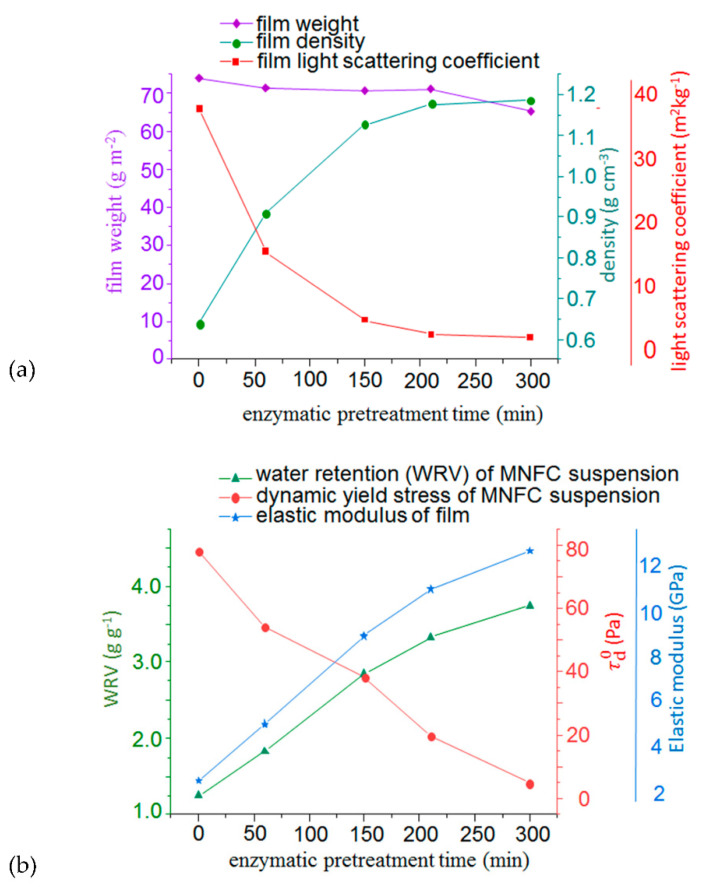
(**a**) Correlation between pulp enzymatic pretreatment hydrolysis time, film density and optical properties, and (**b**) between the water retention and dynamic rheological properties of the MNFC suspension, and the dried film elastic modulus.

**Figure 9 materials-14-03571-f009:**
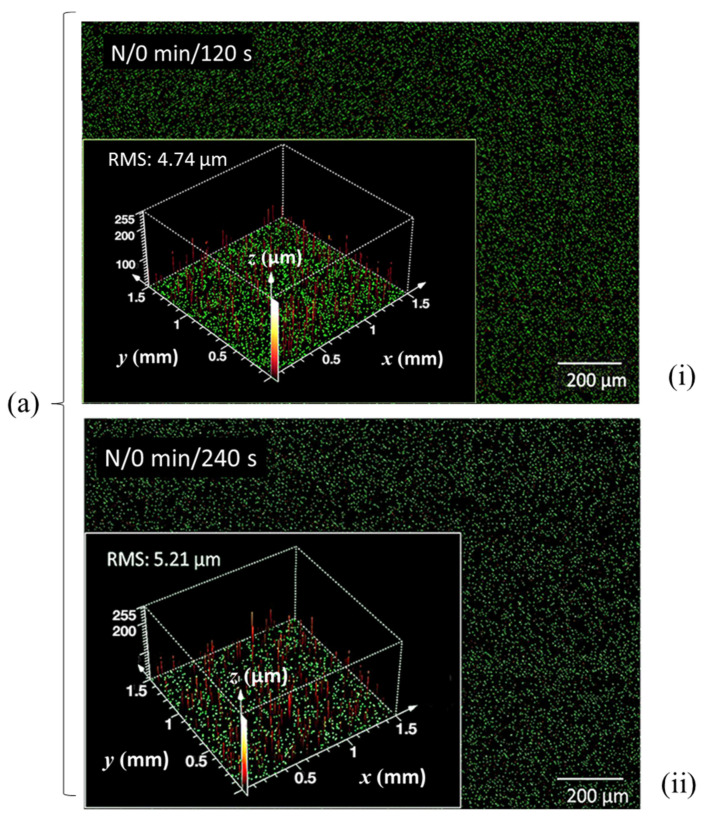
CLSM surface profile plots, showing the microscan as background and contour height spatial distribution in the insert, comparing the effect of enzymatic pretreatment time (**a**) 0 min, (**b**) 60 min, and (**c**) 210 min, producing smoother films before plasma exposure, to the slight roughening effect of N plasma exposure time (**i**) 120 s, and (**ii**) 240 s.

**Figure 10 materials-14-03571-f010:**
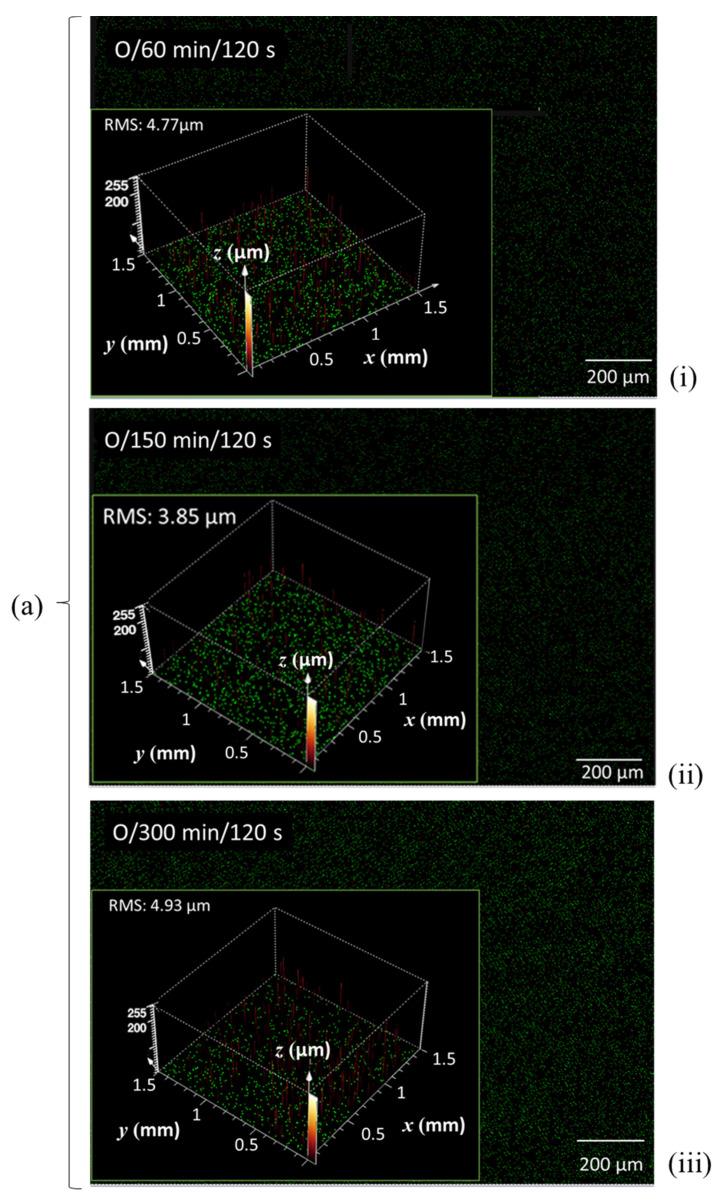
CLSM surface profile plots, showing the microscan as background and contour height spatial distribution in the insert, comparing the effect of enzymatic pretreatment at (**a**) constant O plasma exposure of 120 s (**i**) 60 min, (**ii**) 150 min, and (**iii**) 300 min under enzymatic pretreatment, respectively, and (**b**) comparing (**i**) zero versus (**ii**) 150 min enzymatic pretreatment, both at the longer O plasma exposure time of 240 s.

**Figure 11 materials-14-03571-f011:**
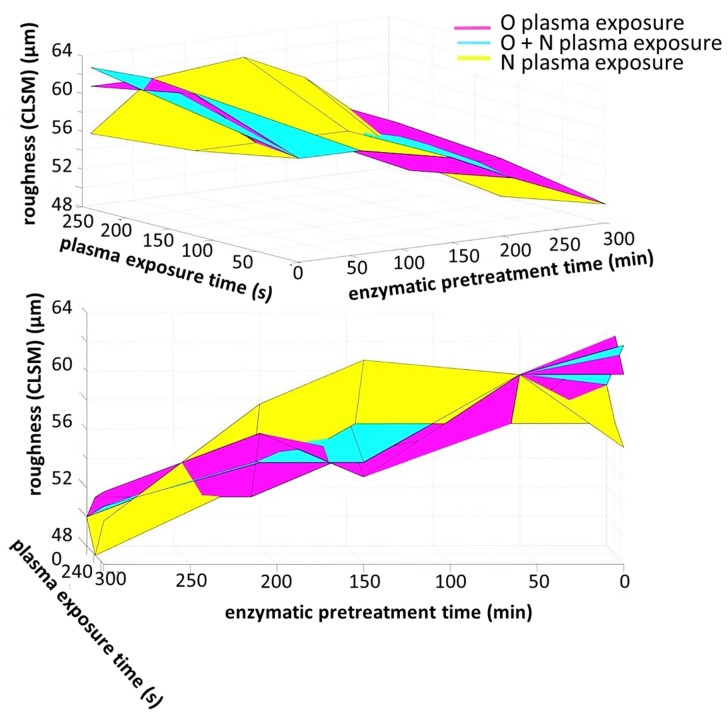
Constructed interpolated 3D relationship surfaces showing the interplay between extending plasma exposure time and enzymatic pretreatment time. The difference between the more aggressive etching of O plasma and N plasma can be easily discerned.

**Figure 12 materials-14-03571-f012:**
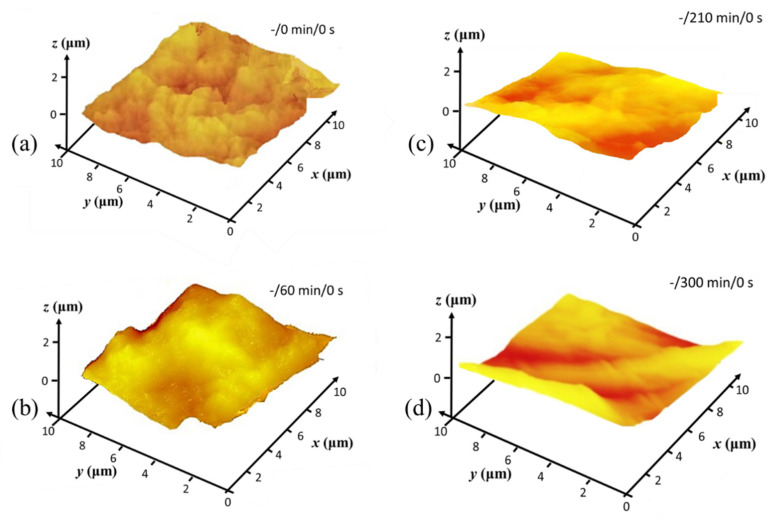
AFM surface profile plots prior to plasma exposure, showing how enzymatic pretreatment generates increased surface nanoscale smoothness (smoother texture between larger scale undulations) during formation of derived MNFC films: (**a**) un-pretreated, (**b**) 60 min, (**c**) 210 min, and (**d**) 300 min of enzymatic hydrolysis pretreatment, respectively.

**Figure 13 materials-14-03571-f013:**
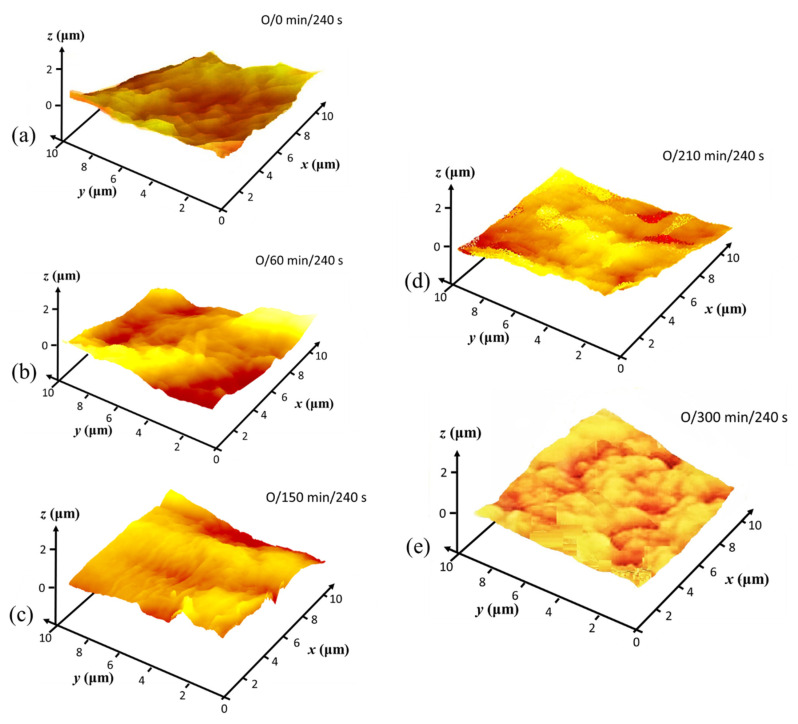
AFM surface profile plots showing the impact of extended oxygen plasma exposure (240 s) in balance with the initial progressively improved smoothness after enzymatic pretreatment: (**a**) 0 min, (**b**) 60 min, (**c**) 150 min, (**d**) 210 min, and (**e**) 300 min pretreatment, respectively.

**Figure 14 materials-14-03571-f014:**
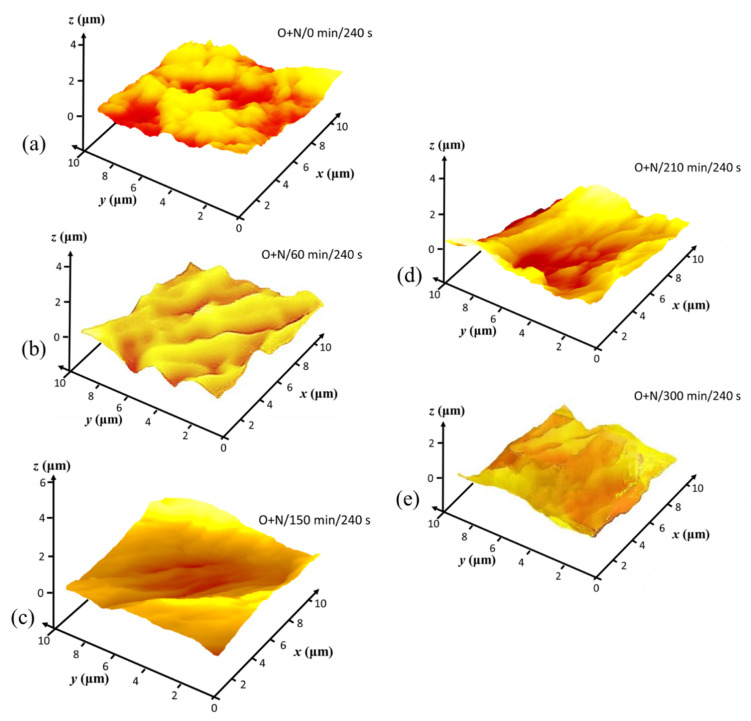
AFM surface profile plots showing the impact of combined 120 s oxygen followed by 120 s nitrogen plasma exposure in balance with the initial progressively improved smoothness after enzymatic pretreatment: (**a**) 0 min, (**b**) 60 min, (**c**) 150 min, (**d**) 210 min, and (**e**) 300 min, respectively.

**Figure 15 materials-14-03571-f015:**
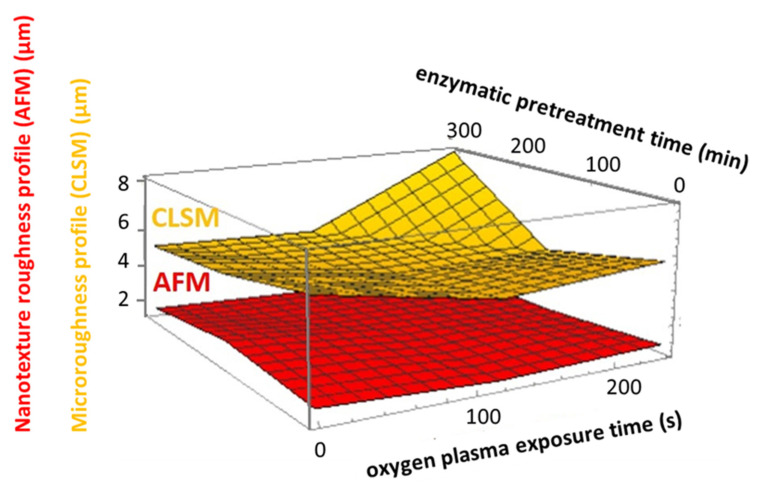
Hierarchical relationship of surface microroughness (CLSM) and nanoscale texture roughness (AFM). The strong etching effect of O plasma (240 s) on the highly hydrolysed enzymatic pretreated MNFC (300 min) countering the original smoothing effect is nicely demonstrated by following the extreme right-hand edge along the enzymatic pretreatment axis.

**Figure 16 materials-14-03571-f016:**
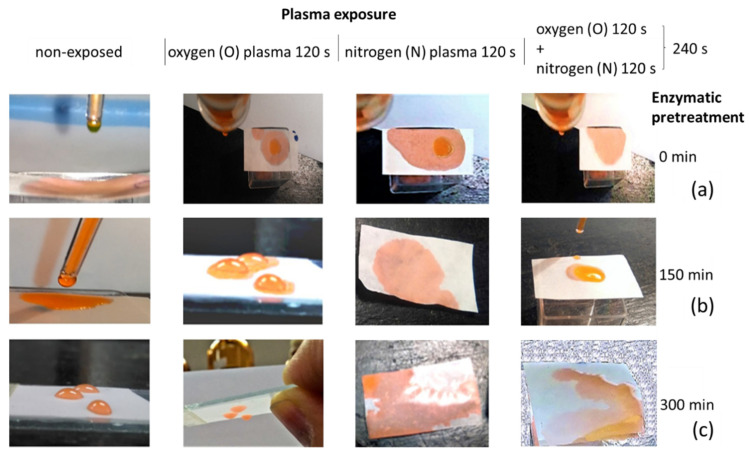
Wetting experiment with drops of dyed water (Eosin dye 0.5%) with different regimes of gas plasma exposure, O and N mono-gas, and two step sequential O + N exposure for films made of MNFC derived different times of enzymatic pretreatment: (**a**) without pretreatment, (**b**) 150 min, and (**c**) 300 min.

**Figure 17 materials-14-03571-f017:**
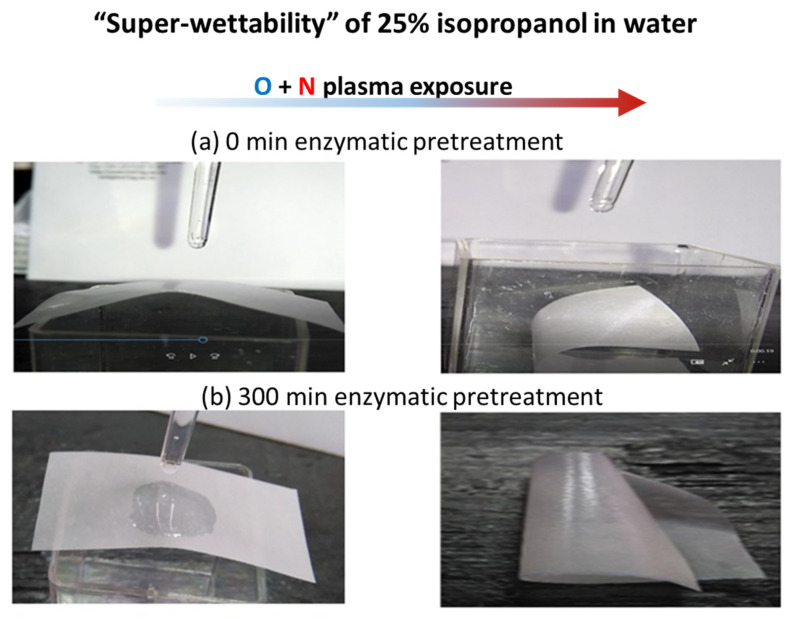
Increased wetting of final N plasma-exposed MNFC films with mix of 25% isopropanol in water isopropanol, showing the effect of plasma radical deposition overcoming the hydrophobic tendency of the more crystalline MNFC derived after long enzymatic pretreatment: (**a**) 0 min, and (**b**) more highly crystalline 300 min fibril enzymatic pretreatment MNFC films, respectively. The arrow left to right shows the transition from O plasma etching only to the combined case of O followed by N plasma exposure.

**Figure 18 materials-14-03571-f018:**
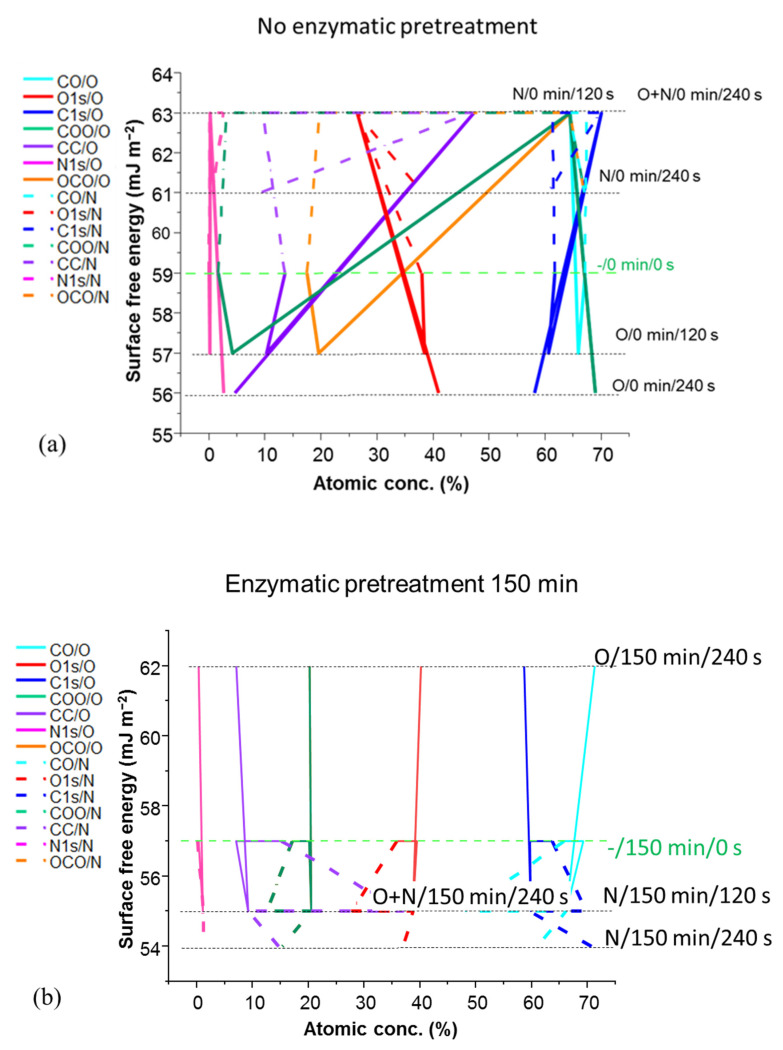
Influence of the O and N plasma exposure on the variously enzymatically pretreated MNFC-derived films by change of chemical properties and accompanying change in effective SFE caused by both induced microroughness and change in atomic concentration at the film surface: (**a**) no enzymatic pretreatment, (**b**) 150 min, (**c**) 210 min, and (**d**) 300 min enzymatic pretreatment, respectively. The joining lines indicate the evolution in SFE as the action of the plasma changes the average atomic moiety concentration, and the data are distributed according to the sample enzymatic pretreatment and plasma exposure histories, with each labelled sample explored marked via the horizontal dotted line recording its given SFE.

**Figure 19 materials-14-03571-f019:**
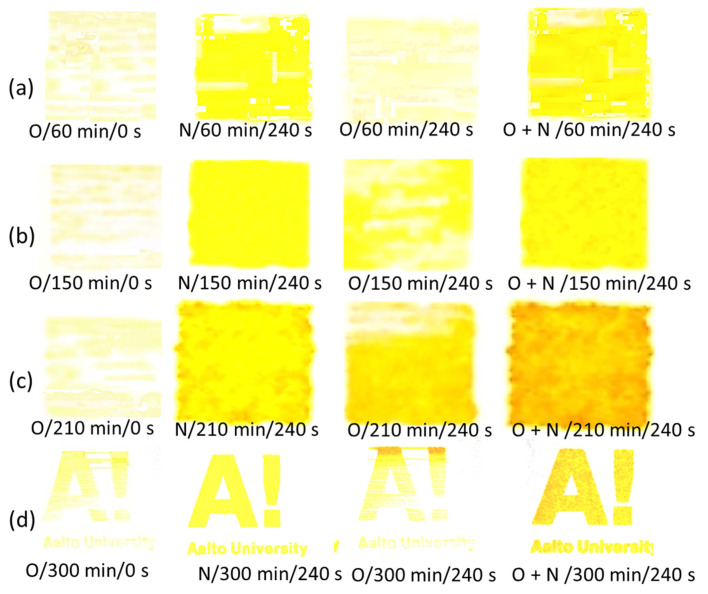
Ink-jet printing using drop on demand (DoD) printing method onto MNFC films, prior to and after O, N, and O + N plasma exposure; (**a**) with 60 min, (**b**) 150 min, (**c**) 210 min, and (**d**) 300 min enzymatic pretreatment.

**Figure 20 materials-14-03571-f020:**
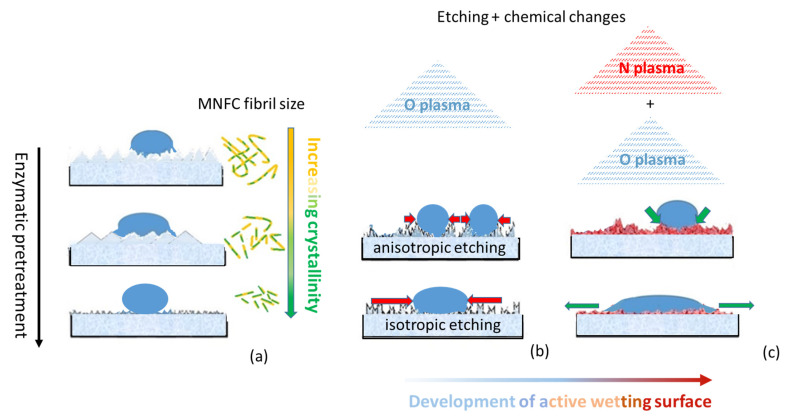
Schematic of how an isotropic mostly crystalline MNFC film can be formed by (**a**) producing the MNFC from enzymatically pretreated fibres, and that (**b**) by judicious exposure to a combination of gas plasma (exemplified by oxygen and nitrogen) (**c**) a desired wettability and surface energy distribution can be generated to provide a new generation of surface active cellulosic materials.

**Table 1 materials-14-03571-t001:** Experimental nomenclature of samples, as used throughout the paper, displaying the enzymatic pretreatment time of pulp (min) and chosen plasma exposure time (s).

Experimental Sequencing of Enzymatic Pretreatment and Plasma Exposure: Plasma Treatment Type/MNFC Enzymatic Pretreatment Time/Plasma Exposure Interval						
MNFC Enzymatic Pretreatment Time (min)	Gas Type Introduced to DBD Plasma Chamber and the Sample Exposure Time (s)					
	No Plasma (-)	Nitrogen		Oxygen		Oxygen (120 s) + Nitrogen (120 s)
(no enzymatic pretreatment) 0	0	120	240	120	240	240
	-/0 min/0 s	N/0 min/120 s	N/0 min/240 s	O/0 min/120 s	O/0 min/240 s	O+N/0 min/240 s
60	-/60 min/0 s	N/60 min/120 s	N/60 min/240 s	O/60 min/120 s	O/60 min/240 s	O+N/60 min/240 s
150	-/150 min/0 s	N/150 min/120 s	N/150 min/240 s	O/150 min/120 s	O/150 min/240 s	O+N/150 min/240 s
210	-/210 min/0 s	N/210 min/120 s	N/210 min/240 s	O/210 min/120 s	O/210 min/240 s	O+N/210 min/240 s
300	-/300 min/0 s	N/300 min/120 s	N/300 min/240 s	O/300 min/120 s	O/300 min/240 s	O+N/300 min/240 s

**Table 2 materials-14-03571-t002:** Water retention and rheological properties of MNFC suspensions (reported as an average of five measurements within a 10% variation).

MNFC Suspension Properties								
Enzymatic Pretreatment Time /min	Water Retention Value (WRV) /cm^3^g^−1^	Dynamic Drainage Jar MNFC Content in Filtrate /*w/w*%	Dynamic Yield Point ($\tau_{d}^{0}$) /Pa	Static Yield Point ($\tau_{s}^{0}$) /Pa	Consistency Coefficient (*k*) /Pa.s	Complex Consistency Coefficient (*k**) /Pa s*^n^*^*^	Shear Thinning Coefficient |(1 − *n*)|	Complex Shear Thinning Coefficient |(1 − *n**)|
0	1.25	95.8	334.12	478.16	431.23	482.46	0.82	0. 86
60	1.83	78.6	74.23	81.23	139.65	163.51	0.81	0.87
150	2.85	25.4	53.34	88.34	57.23	73.47	0.84	0.88
210	3.33	10.7	7.64	11.64	19.67	22.67	0.86	0.93
300	3.34	1.3	4.74	6.74	5.45	7.26	0.91	0.95

**Table 3 materials-14-03571-t003:** RMS microroughness values from CLSM scans of selected plasma-exposed MNFC films, showing the impact of N and O plasma exposure, respectively. Selection was based on samples showing the greatest property differences between them.

CLSM Scan														
**Enzymatic pretreatment time** **(min)**	0		60		210		0		60		150		300	
	**N plasma**						**O plasma**							
**Plasma exposure time** **(s)**	120	240	120	240	120	240	120	240	120	240	120	240	120	240
**RMS microrough-ness** **(μm)**	4.74	5.21	4.46	4.71	3.19	3.65	-	5.63	4.77	-	3.85	4.11	4.93	-

**Table 4 materials-14-03571-t004:** Derived surface free energy (SFE) values of MNFC films displayed over the range of both enzymatic hydrolysis pretreatment and plasma gas type exposure time (see also Table 1).

Plasma	Plasma Exposure Time (s)	Duration of Enzymatic Pretreatment (min)				
		0	60	150	210	300
		Surface Free Energy (SFE) (mJ m^−2^)				
No exposure	0	59	59	57	54	50
N (nitrogen mono-gas)	120	63	57	55	52	52
	240	61	63	59	57	53
O (oxygen mono-gas)	120	57	57	57	53	48
	240	56	61	62	59	51
O + N (sequential oxygen followed by nitrogen)	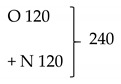	63	61	55	54	52

## Data Availability

The data presented in this study are available on request from the corresponding author.

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
