# Peer review of "Iso- and Anisotropic Etching of Micro Nanofibrillated Cellulose Films by Sequential Oxygen and Nitrogen Gas Plasma Exposure for Tunable Wettability on Crystalline and Amorphous Regions"

_materials, 2021, doi:10.3390/ma14133571_

Round 1

Reviewer 1 Report

The authors have investigated the etching of micro nanofibrillated cellulose films by sequential oxygen and nitrogen gas plasma exposure for tunable wettability on crystalline and amorphous regions. The authors have obtained some good results in this work. However, some minor corrections still needed. The manuscript title is too long. It should be shortened. The introduction section is also 3 page long. It sounds like a review article. Please reduce it to 1 and half, if possible. The discussion of Fig. 3 needs to be improved. The exposure time scale in Fig. 11 should be re-scaled for better 3D view. The mechanism given in Fig. 20 needs more elaboration.

Author Response

All reviewer comments have been responded to and respective changes to text, Tables and Figures undertaken.

2. Complete text revision by a native scientific English speaker has been applied.

 Therefore, all changes are shown in blue text, including Figure captions, which, when agreed, can be readily turned to black.

Reviewer 2 Report

The paper introduced the novel method to produce the Iso- and anisotropic etching of micro nanofibrillated cellulose films. The important properties were tested and evaluated. It provided useful information. However, some content need to be improved.

  1. Figure 2b have provided a clear figure. The (a) part could be deleted.
  2. Line 346, some type mistakes.
  3. Table 1. “Experimental configuration of samples displaying the enzymatic pretreatment time of pulp and chosen plasma 352 exposure time.” Could be improved. Some words, such as /0s, /120s, /240s could be deleted. For example, -/0 min/0 s replaced as -/0min, -/30 min/0 s replaced as -/30 min.
  4. Line 398, please improve the quality. Same as line 410.
  5. Table 2. Dewatering and rheological parameter properties of MNFC suspensions.

All properties were one replicate or several replicates. If they were measured several time, please indicated as mean and standard deviation.

  1. Please express the results in Figure 8 as three parts. Each part have two properties only.
  2. The quality of Figure 18 could be improved. It is too complicated to be understand.

Author Response

The review process has resulted in a truly extensive, fundamental overhaul of the manuscript:

1. Rreviewer  2comments have been responded to and respective changes to text, Tables and Figures undertaken.

The result is that the multiple changes were too cumbersome using tracking and impossible to follow. Therefore, all changes are shown in blue text, including Figure captions, which, when agreed, can be readily turned to black.

Reviewer 3 Report

Dear authors,

Comments and instructions may be found in the attached Word document.

Best regards

Author Response

We appreciate the detail in which Reviewer #3 has studied the manuscript text and provided a lengthy Table of suggested editorial corrections needed. Many of these have now been rectified according to the complete manuscript linguistic check. The remaining have been tackled separately. Each item is commented in turn in the Table (please note that line numbers are not correlating now after the multiple changes made to accommodate the previous review changes above).

1. All reviewer comments have been responded to and respective changes to text, Tables and Figures undertaken.

2. Complete text revision by a native scientific English speaker has been applied.

The result is that the multiple changes were too cumbersome using tracking and impossible to follow. Therefore, all changes are shown in blue text, including Figure captions, which, when agreed, can be readily turned to black.

We thank you for your consideration of this re-submission, and thank also the reviewers for their extensive input.

Round 2

Reviewer 3 Report

Thank you for revising the paper.